# Use of Secondary Metabolites Profiling and Antioxidant Activity to Unravel the Differences between Two Species of Nettle

**DOI:** 10.3390/plants12183233

**Published:** 2023-09-11

**Authors:** Julia Baumli, Norbert Antal, Dorina Casoni, Claudia Cimpoiu

**Affiliations:** 1Faculty of Chemistry and Chemical Engineering, Babes-Bolyai University, 11 Arany Janos, 400028 Cluj-Napoca, Romania; julia_baumli76@yahoo.ro (J.B.); anorbertcsaba@yahoo.com (N.A.); dorina.casoni@ubbcluj.ro (D.C.); 2Research Center for Advanced Chemical Analysis, Instrumentation and Chemometrics, 11 Arany Janos, 400028 Cluj-Napoca, Romania

**Keywords:** extraction optimization, Box–Behnken method, iridoids, secondary metabolites profiling, antioxidant activity, thin-layer chromatography, RP-UHPLC, *Lamium album*, *Urtica dioica*

## Abstract

In recent years, the interest in natural remedies has increased, so it is important to analyze the plants widely distributed in nature but whose composition is little known. The main objective of the present work is to obtain information based on the profiles of secondary metabolites and antioxidant activity in *Lamium album*, a very widespread but little studied plant, with the aim of revealing the differences compared to *Urtica dioica*. First, the optimization of enzymatic extraction assisted by ultrasound was carried out by the Box–Behnken method. The optimized parameters were: concentration of the enzyme—3.3% cellulase, temperature—55 °C, and the extraction time—40.00 min. The efficiency was estimated based on the content of iridoids, the main class of secondary metabolites from *Lamium album.* Second, the secondary metabolites profiles of the nettle extracts were obtained by thin-layer chromatography using both normal and reverse phases and by RP-UHPLC. The antioxidant activity was evaluated using DPPH and ABTS^+^ radicals. The obtained results revealed significant differences between the two nettle species, both in terms of the phytochemical compounds, as well as the antioxidant activity, confirming the fact that *Lamium album* has a high potential to be used in phytomedicine.

## 1. Introduction

Since ancient times, people have used plants or plant extracts as medical treatments for various pathologies. Plant extracts are used due to their high concentration of bioactive compounds, which are substances responsible for certain physiological activities carried out in the body of living organisms. A wide variety of plants have been highly researched by scientists with the aim of curing various diseases naturally without using synthetic drugs. Along with the development of technology, our evolution also takes place, which results in a better understanding of the way that bioactive compounds from plants act on organisms; new applications of extracts are discovered and thus remedies are found for diseases that have been considered to be incurable until now. Unfortunately, the plant materials used for herbal preparations differ in terms of quality and variability, and the quality assurance requirements for herbal products are not as strict as those in the pharmaceutical industry. For these reasons, increased efforts are being made to effectively regulate the medicinal plant industry, which has led to the need to develop accurate and reliable methods for the quality control and standardization of plant material. For the identification of plant materials, specifically their extracts, three approaches are used, namely: the marker approach, the multi-component approach and the model or fingerprint approach [1]. The use of fingerprints, which are chemical profiles of different constituents in plant material, is widely accepted as an effective way to describe the complexity of the components present in a plant because, in most cases, a comprehensive chemical profile is generally unavailable [2,3,4].

*Lamium album* and *Urtica dioica* are not related, even if they are similar in appearance (both being species of nettle). Although *L. album* and *U. dioica* are species of nettle, their therapeutic effects can be the same [5], but they can also be opposite [6] due to the secondary metabolites contained; in some cases, the use of one of the plants may even be contraindicated. *Lamium album* is traditionally called the “white dead nettle” or “non-stinging nettle”. This plant is known for its usefulness in both the culinary and in the medical environments [7]. *Lamium album* is a perennial herbaceous plant that belongs to the family *Lamiacea*, genus *Lamium*, which contains about forty species distributed almost all over the world. The plant grows to a height between 50 and 75 cm. The leaves are elongated, triangular, with a rounded base, serrated edge and covered with soft bristles. The flowers of *Lamium album* are white, double-lipped, and positioned at the top of the stem in a collar of six to twelve flowers (Figure 1). The growing area is in the vicinity of water and forests, but it can also be found in lowland areas. *Lamium album* is called a “nettle” because phenotypically it looks similar to the common nettle, *Urtica dioica* [8]. However, the bristles on the leaves of the *Lamium album* plant are not prickly, making it harmless, as if it were “dead”. *Lamium album* contains many biologically active substances, such as: iridoids, flavonoids, phenolic and fatty acids, polysaccharides and essential oils. It is known to be effective in chronic bronchitis or pharyngitis due to its mucolytic and antispasmodic activity. In addition, the plant extract is also applied in skin inflammations and alimentary or urinary tract diseases, showing beneficial effects for the whole body by stimulating the removal of harmful products of metabolism [9]. *Lamium album* is an edible plant that is eaten raw or cooked, especially in and around the Mediterranean area. It can also be used in food in the form of tea infusions and as the main component of food supplements, due to its pharmacological effects (anti-inflammatory, astringent, antiseptic, antiviral, antimicrobial, antioxidant, anticarcinogenic, cytoprotective, wound healing) [6,10]. Among the most prominent secondary metabolites in *Lamium* species are iridoids, with a glycosidic structure. Iridoids are compounds characterized by a pyranoid skeleton, often referred to as the iridan skeleton, and are found in plants most often as glycosides but may also exist in a dimerized or nonglycosidic form [11]. Iridoids possess a wide range of pharmacological activities, including anticancer, anti-inflammatory, antifungal and antibacterial activities.

*Urtica dioica* is a perennial herbaceous species belonging to the *Urtica* genus of the *Urticaceae* family. It is special because it has certain prickly bristles on its leaves with a self-protective role. *Urtica dioica* is much more widespread than *Lamium album* and is widely used in medicine, thanks to its beneficial effects on living organisms [12].

*Urtica dioica* is widely found in Europe, Asia and Africa and it is also used in medicine due to its beneficial therapeutic effects. Numerous bioactive chemical components, including flavonoids, phenolic acids, amino acids, carotenoids and fatty acids, have been identified from the stinging nettle to date that provide plant nutritional benefits and anti-inflammatory and antioxidant capacities [13].

Obtaining the chromatographic fingerprints that represent the phytoequivalence of a plant depends, among other things, on the efficient extraction of the compounds present in the plant material. Among the innovative extraction techniques, ultrasound-assisted enzymatic extraction (UAEE) has proven to be a fast, efficient, simple and inexpensive method that can provide high reproducibility. In order to obtain satisfactory yields in any extraction process, the optimization of its parameters such as the concentrations of enzymes, solvent composition, temperature, time and solvent-to-solid ratio is of particular importance. There are various methods of experimental design, each method being based on the same steps to follow: planning, executing the experiment and analyzing the collected data to be able to draw the appropriate conclusions [14]. The Box–Behnken model combined with the response surface methodology (RSM) can be used to optimize complex processes when the chosen parameters have a combined effect on the response values. This approach has the main advantage of significantly reducing the number of experimental trials needed for evaluating multiple parameters and their interactions and generating a mathematical model to find the optimal values [15].

Therefore, the main objective of the present work is to evaluate the effects of different working parameters on UAEE for iridoids and to obtain information based on the chromatographic profiles of secondary metabolites and antioxidant activity to differentiate *Lamium album* and *Urtica dioica* plant materials.

## 2. Results and Discussion

### 2.1. Optimization of the UAEE

The optimization of time, temperature and cellulase concentration on the UAEE of secondary metabolites from *Lamium album* was carried out. First, single factor experiments were performed by simultaneously varying a single parameter, the others being kept constant at the average value corresponding to the analyzed domain. The change in the total iridoids content (TIC) was monitored as the outcome of extraction. The importance of this step of reducing the work intervals is that, by optimizing the process, much better results can be achieved than if a wide field were considered.

Thus, the effect of cellulase amounts relative to the plant material at various quantities (between 1.5% and 3.5%) on extraction of TIC was studied, keeping the working temperature at 50 °C and the extraction time at 25 min (Figure 2a). In the case of the extraction time effect on the extraction of TIC, the concentration of cellulase was kept at 2.5% and the temperature was maintained at 50 °C (Figure 2b); for the temperature effect, the enzyme concentration was 2.5% and the extraction time was 30 min (Figure 2c).

Figure 2a shows that at cellulase concentrations above 3%, the TIC value began to decrease, which indicated that the amount of cellulase of 3.0% relative to the plant material is sufficient to completely hydrolyze the cell walls of the plant and to release the iridoids. Regarding the extraction time, from Figure 2b it can be seen that although the increase in the extraction time would be expected to have a positive effect on the efficiency, this does not happen, with TIC decreasing at an extraction time of 40 min. This could be explained by the fact that with the excessive extension of the extraction time, the degradation of the extracted iridoids by hydrolysis or oxidation could occur. Regarding the extraction temperature, it can be seen from Figure 2c that the TIC increases with the increase in temperature up to 60 °C and then decreases; this is probably due to the thermal degradation or oxidation of the iridoids, as well as the denaturation of cellulase. These results are in agreement with the results obtained by other authors for the extraction of secondary metabolites from plants [15].

Following the analysis of Figure 2, new narrower working intervals were obtained, where the maximum values obtained from the TIC are located. These working intervals for the application of the optimization method are presented in Table 1.

The matrix of the experimental design corresponding to the Box–Behnken model following the experiments and the obtained response variable is presented in Table 2. The TIC values rank from 0.294 µg of aucubin/µL for Experiment 3 (20 min, 3.5% cellulase and 60 °C) and 0.530 µg of aucubin/µL for Experiment 6 (40 min, 3.5% cellulase and 50 °C).

The obtained polynomial equation corresponding to the Box–Behnken model takes the following form after estimating the coefficients:Y = 0.5164 + 0.07573A + 0.02215B − 0.01287C − 0.0642A^2^ − 0.0731B^2^ − 0.0977C^2^ + 0.0403AB − 0.0633AC − 0.0339BC(1)

The results obtained after carrying out the 15 planned experiments based on the Box–Behnken model comprise three independent variables and three levels, as well as three replicates at the central point. The resulting equation shows the relationship between the TIC value of the extracts and the extraction parameters. To test the adequacy of the model and the significance of the coefficients, the analysis of variance (ANOVA) test was used. The results obtained after applying the analysis algorithm are summarized in Table 3.

The following conclusions can be reached from the ANOVA results: the adequacy of the model is high (the F value (30.02) is higher than its critical value (2.6458)); the *p* value of 0.001 of the model against 0.05 (imposed limit at the 95% confidence level) confirms the adequacy of the model; the significant variables for the model (*p* < 0.05) are A, B, A^2^, B^2^, C^2^, AB, AC, BC; variable C is not significant for the model in its linear form, but in interaction with other variables or in quadratic form it influences the model; the high value of R^2^ (0.9818) indicates the high adequacy of the model, with less than 1% of the total variation not being explained by this model.

In Figure 3, the response surfaces of the combinations of two independent variables and the dependent variable are represented. From these three representations, it can be deduced that the analyzed interval includes the maximum point, a fact that emerges from the curved shape of the representations that indicates an increase up to a point and then a decrease in the value of the response.

The response surface plots (a) and (b) indicate that the required maximum is found closer to the upper limit of the search interval for the extraction time that appears in both plots. Regarding the plot (c), it can be observed that the enzyme activity decreases with increasing temperature, the enzyme activity maximum being located around 55–60 °C, which is in agreement with the data provided by the enzyme’s manufacturer.

The optimization of the polynomial equation in order to obtain the maximum was carried out in the Matlab program with the function fmincon and the following optimal values were obtained: A* = 0.9952; B* = 0.5375; C* = −0.4815. The optimal values were obtained by linear interpolation using the interp1 function and are as follows: extraction time—39.9523 min; enzyme concentration—3.2687% mass percentage of the amount of plant material; and extraction temperature—55.1849 °C.

Following the estimation of the optimum conditions for the extraction, the experimental verification of the optimum took place. The model predicted value for the optimal parameter values is 0.563 µg/µL and the value obtained experimentally is 0.532 µg/µL. The deviation of the obtained value from the predicted value was calculated to be 5.8%. This value falls between the imposed limits of ±10% and is due to the impossibility of respecting exactly the optimum parameters; thus, they are slightly modified, being time—40 min, enzyme concentration—3.3%, and temperature—55 °C.

### 2.2. Determination of the TIC in the Extracts

Five extracts (four extracts from *Lamium album* and one extract from *Urtica dioica*) were prepared using the UAEE and selected optimal parameters. The TIC in each extract was determined spectrophotometrically and the results are presented in Figure 4.

The TIC in the *U. dioica* extract (0.152 µg/µL) is much lower than in the extracts from *L. album* (medium value—0.445 µg/µL), confirming the fact that the iridoids are a class of secondary metabolites specific to the *Lamium album* plant. If the TIC for the extracts of *L. album* are compared, it is found that the values are close, with the extract E1 being similar to E5 (*p* < 0.05). It can also be seen that E2 contains the greatest TIC, probably due to the fact that the plant was more mature and flowery.

### 2.3. Chromatographic Secondary Metabolites Profiling

The identification of secondary metabolites in each sample is important in order to establish some differences, but this is not absolutely necessary because obtaining fingerprints can allow fast and safe identification of the samples, which has been demonstrated by various studies [16,17,18]. For this reason, the purpose of the chromatographic analyses is to obtain the secondary metabolites profiles of the two plants that can distinguish the difference between them, fingerprinting often being recommended even by regulatory agencies as a basis for the proper identification of plant products [19].

#### 2.3.1. Thin-Layer Chromatography (TLC)

TLC has become a powerful analytical method for analyzing complex samples with advances in instrumentation and the use of modern digital image evaluation techniques that can combine different images of the same fingerprint for a reliable evaluation. Thus, TLC was included as a method of identification of medicinal plants based on a chromatographic imprint in most monographs in the pharmacopoeia [1]. Interpretation of chromatographic data is performed by combining TLC with image analysis (IA) and chemometric methods.

The TLC profiles of the secondary metabolites are obtained on both the normal phase and reversed phase. Consequently, the extracts of *Lamium album* and *Urtica dioica* were investigated for the presence of polyphenols and iridoids, as well as for the evaluation of antioxidant activity. The complexity of the chromatographic fingerprints required multivariate analysis to evaluate the similarity/dissimilarity in the fingerprints so that useful information could be extracted to reveal the samples’ differentiation.

The TLC profiles of polyphenols were obtained in UV light before and after derivatization with NP/PEG. In order to verify the presence of possible polyphenols, certain common standards that were previously identified in *Lamium album* [20] were applied to the plate, together with extracts, namely: p-coumaric acid, vanillic acid, rutin, caffeic acid, quercetin, ferulic acid, gallic acid, chlorogenic acid and protocatechuic acid.

In Figure 5, the images of a normal-phase chromatographic plate eluted with the mobile phase consisting of ethyl acetate–methanol–water (7.7:1.5:0.8 *v*/*v*/*v*) and the reversed-phase chromatographic plate (RP-18) eluted with a mixture of formic acid (0.1%)–acetonitrile (5:5 *v*/*v*) can be seen before derivatization, at 254 nm and 366 nm.

From the analysis of these separations, as an initial hypothesis, the presence of polyphenols can be estimated as follows: p-coumaric acid (S1) is found in E2; vanillic acid (S2) is found in E1; rutin (S3) is found in E1, E2, E4 and E5; caffeic acid (S4) appears to be present in E3 but requires further confirmation; quercetin (S5) does not appear to be present in any of the extracts; ferulic acid (S6) seems to be present in E4; gallic acid (S7) looks to be present in traces in all extracts but needs further verification; chlorogenic acid (S8) looks as if it is present in all extracts but in a higher concentration in E4; and protocatechuic acid (S9) cannot be confirmed in any of the extracts.

An intensification of fluorescence is seen after the spraying of the plates with NP and PEG (Figure 6), leading to the following findings: the presence of rutin (S3) is confirmed in all extracts, in higher concentrations in E1, E2 and E5 than in E3 and E4; the extract E3 does not contain caffeic acid (S4); quercetin (S5) and gallic acid (S7) are confirmed not to be found in any of the extracts; the presence of chlorogenic acid (S8) in all extracts is confirmed, being in the highest concentration in E4; and protocatechuic acid (S9) is not present in the analyzed extracts.

Data from the TLC fingerprints of separated polyphenols (obtained using detection under UV at 254 nm and 366 nm before and after derivatization of the TLC plates, respectively) also revealed a clear differentiation of the *U. dioica* and *L. album* samples based on the score plot of the first two PCs (accounting for 75.05% in case of silica gel plates and 80.64% in case of RP-18 plates) (Figure 7).

A stronger contribution for samples’ classification (PC1 loadings > 0.9, Appendix A) was revealed from certain separated polyphenols by both UV 254 nm (R_f_ values 0.1–0.3; 0.7 and 0.8) and UV 366 nm detection (R_f_ values 0.4; 0.5 and 0.7–0.8 without pulverization and 0.1–0.2 and 0.3–0.4 after pulverization).

The data from TLC profiles of polyphenols on RP-18 plates revealed a good differentiation between *L. album* samples collected from different regions in Romania. According to PC1 (PC1 loadings > 0.9, Appendix A), compounds separated at R_f_ values 0.8–0.9 and detected at UV-366 nm have the most significant contribution for their differentiation. Both types of chromatographic plates highlighted that the sample collected from a traditional producer (E5) has a different composition in polyphenols comparative with the samples from spontaneous flora.

The presence of iridoids in the extracts was revealed by spraying the chromatographic plates with specific reagents for iridoids. For this purpose, after separating the target compounds on both silica gel and RP-18, the plates were sprayed with the Ehrlich reagent and anisaldehyde (Figure 8).

The images of the plates indicate the presence of iridoids in the analyzed extracts as blue-grey or blue-violet bands. Aucubin (S10) was used as the control compound to test for the proper activity of the detection reagent and to reveal the color of the bands containing iridoids. From these images, it appears that aucubin cannot be found in these extracts and few colored bands of variable intensity are exhibited. Also, these results sustain the spectrophotometric analyses that revealed a significantly lower content of total iridoids in the *Urtica dioica* sample (E4) comparative to *Lamium album* samples (E1, E2, E3 and E5).

The PCA analysis also confirm that *Urtica dioica* (E4) is clearly differentiated from *Lamium album* samples (E1–E3, E5) by the TLC iridoids profiles (Figure 9).

The score plots of the first two PCs accounting for 86.32% in the case of silica gel plates and 92.24% in case of RP-18 plates reveal a similar composition in iridoids for samples collected from Satu Mare and Cluj-Napoca (E1 and E2). The use of silica gel plates revealed that sample from traditional producer (E5) has a different composition in iridoids compared with the *L. album* samples collected from spontaneous flora.

Iridoids separated on silica gel plates at R_f_ values of around 0.41–0.43 detected with anisaldehyde and at R_f_ values of about 0.11–0.52 detected with the Ehrlich reagent have the strongest contribution (PC1 loadings > 0.90, Appendix A) in differentiating *U. dioica* from *L. album* samples. According to the PC2 loadings, the iridoids separated at an R_f_ of about 0.1 and detected using anisaldehyde and at R_f_ values of 0.5–0.7 and detected using the Ehrlich reagent clearly differentiate the *L. album* samples (PC2 loadings > 0.90, Appendix A) collected from different regions in Romania. The iridoids separated on RP-18 plates at R_f_ values between 0.5 and 0.7 and detected using anisaldehyde and at an R_f_ between 0.3 and 0.4, 0.5 and 0.6, and 0.7 and 0.8, respectively, detected using the Ehrlich reagent clearly differentiate (PC1 loadings > 0.9) *U. dioica* from *L. album* samples.

#### 2.3.2. Liquid Chromatography

HPLC has been widely used for the fingerprints because it can successfully separate the different constituents of the extract [21].

The UPLC chromatograms recorded at 210 nm provided more detailed profiles and characterizations of the samples regarding iridoids (Figure 10).

It can be seen that the extract of *U. dioica* (E4) shows a much lower number of peaks compared to the other extracts, showing that iridoids do not have as high prominence as in *Lamium album*. As for the differences observed in the chromatograms of *L. album* extracts (E1–E3 and E5), the variation in the concentration of certain compounds is observed depending on the place of origin. In this way, the concentration of the compound corresponding to a retention time of 5.08 min, 6.43 min, 6.70 min, 13.58 min and 15.59 min could distinguish between the extracts of *L. album*. In general, the highest concentrations of iridoids are observed in the E2 extract, as in the case of the TLC profile. According to the retention times and comparing with data from the literature [22], the following iridoids can be identified in the *L. album* extracts: lamalbid (RT—1.84 min), caryoptoside (RT—3.34 min), lamuisid A (RT—10.97 min) and tiliroside (RT—22.55).

According to the PCA results from the UHPLC chromatogram, the plot of the first two PCs includes 85.19% of the total variance (PC1 = 70.07%, PC2 = 15.12%). From the score plot of the first two PCs (Figure 11), we can see that PCA clearly distinguishes *L. album* samples from *U. dioica,* indicating that the two investigated species are relatively different in chemical composition.

In addition, it can be observed that the *L. album* samples collected from spontaneous flora in Romania (E1–E3) are similar in their composition (grouped with close PC1 values) and well separated from samples collected from the controlled cultures of the producer (E5, grouped with higher values for PC1). According to the loadings values from varimax rotation analysis (PC1 loadings > 0.90, Appendix A), the highest contribution in differentiating *L. album* and *U. dioica* species is associated with constituents separated from the 1.8–3.5 min, 6.5–6.6 min, 9.9–10 min, 12.9–13.2 min, 18.1–18.6 min, 20.1–22.3 min and 22.7–23.8 min retention time intervals, respectively (Appendix A). Samples collected from Cluj-Napoca (E2) and picked during the flowering period are also relatively different in chemical composition according to the loadings of PC2 that are associated with a strong contribution (PC2 loadings > 0.90) of components separated from the 4.5–5.2 min, 10.8–11.1 min, 13.3–13.8 min and 15.3–15.9 min retention time intervals, respectively (Appendix A).

### 2.4. Antioxidant Activity

To evaluate the antioxidant activity of the extracts, the TLC plates were immersed after separation both in ABTS•^+^ solution and DPPH• solution. From the images of the plate (Figure 12), the appearance of the yellow spots on a green or violet background can be observed indicating the presence of antioxidant compounds.

Extracts of *Lamium album* (E1-E3 and E5) are observed to show much a greater antioxidant activity than the extract of *Urtica dioica* (E4), revealing another difference between these two species of nettle. It can be identified that, in the case of reversed-phase plates, the antioxidant compounds are concentrated at higher retention factor values than in the case of silica gel plates. In order to better evaluate the antioxidant potential of extracts, the antioxidant activity, calculated as the sum of the all-peaks area was determined for each of the analyzed extracts after the background subtraction [18] and the results are presented in Table 4.

The differences between antioxidant activity values appear to be due to the sensitivity of the two radicals to the hydrophilicity and lipophilicity of the compounds. ABTS•^+^ is much more sensitive in the presence of hydrophilic antioxidants, while DPPH• is in the presence of lipophilic ones. As can be seen, although statistically no significant difference was found (*p* > 0.05), the values obtained with ABTS•^+^ are slightly lower than those with DPPH•, which may indicate the presence of more lipophilic compounds than hydrophilic ones in the extracts. Instead, regardless of the radical used, a significant difference (*p* < 0.05) was found between the values obtained on the silica gel plates and the RP-18 ones, with higher values being obtained on the latter. This could be due to the blocking of certain reactive groups by the bonds formed between the compounds and the stationary phase in the case of silica gel, but this statement requires further research.

## 3. Materials and Methods

### 3.1. Chemical Reagents

All used reagents are of analytical purity as specified by suppliers. Ethanol (96%), glacial acetic acid (≥99.7%), sulfuric acid (95.0–98.0%), hydrochloric acid (37%), copper sulfate (≥98.0%), anisaldehyde (≥98.0%), 2,2′-Azino-bis (3-ethylbenzothiazoline-6-sulfonic acid) diammonium salt (ABTS, ≥98.0%), 1,1-Diphenyl-2-picrylhydrazyl radical, (DPPH•, ≥98.0%), K_2_S_2_O_8_ (≥99.0%), para-(dimethylamino)benzaldehyde for TLC derivatization, polyethylene glycol 400 (macrogol, tested according to Ph. Eur.), 2-aminoethyl diphenylborinate (97.0%), p-coumaric acid (≥98.0%), vanillic acid (≥97.0%), caffeic acid (≥98.0%), ferulic acid (≥97.0%), gallic acid (≥98.0%), chlorogenic acid (≥95.0%), protocatechuic acid (≥97.0%), rutin (≥94.0%), quercetin (≥95.0%) and aucubin (≥98.0%) were purchased from Merck (Darmstadt, Germany). Cellulase was purchased from Pangbo Enzyme (Guangxi, China). The solvents used for chromatographic analysis were purchased from Millipore, Bedford, USA, and have the following purities: methanol ( ≥99.8%), ethyl acetate (≥99.5%), acetonitrile (≥99.9%) and formic acid (≥98.0%).

### 3.2. Plant Material

In this paper, both *Lamium album* and *Urtica dioica* are analyzed to highlight the differences between the two species. The plants come from different areas in the north-west of Romania: from Satu Mare county (in April 2022), from Dej (Cluj county in May 2022), from the city of Cluj-Napoca (in June 2022). Also, *Lamium album* tea and *Urtica dioica* tea were manufactured in 2022 from the culture of producers and were bought from a specialized market. The whole fresh plants were dried in the dark at room temperature (22–24 °C), in a ventilated space and then the material was ground in order to obtain a fine and homogeneous powder with granules of equal size.

### 3.3. Optimization of Extraction

UAEE experiments were performed in Emmi-20 HC (EMAG, Germany ) ultrasound bath with time and temperature controller, using 0.5 g of ground plant material and 5 mL 55% ethanol as a solvent and the enzyme concentration, the extraction time and the temperature being fixed by experiment design. After extraction, the supernatants were separated from the solid residue by centrifugation at 6000 rpm for 5 min using a Centurion Scientific centrifuge C2006 (Centurion Scientific Limited, Bosham, UK) and then used for determination of the total iridoids content (TIC).

In order to fix the independent variables for the optimization of this system, multiple extractions were performed at different values of the main parameters that can have a significant influence on the extraction, namely the enzyme concentration (0.5–5% cellulase respecting to the amount of plant), extraction time (10–50 min) and temperature (30–70 °C). Carrying out the experiments with the variation in a certain parameter took place with the constant maintenance of the other variables at an average value.

Following the reduction in the search intervals for the maximum values of TIC, the Box–Behnken method was chosen to plan the experiments. The matrix of experimental levels was obtained, having a total of 15 experiments. The extractions were performed at the values of the variables indicated in the matrix. Finally, the values of the response variable were obtained.

The regression equation of the Box–Behnken model is a polynomial function, which has the following form for a three-factor, three-level system:Y = c_0_ + c_1 × 1_ + c_2_X_2_ + c_3_X_3_ + c_4_X_1_^2^ + c_5_X_2_^2^ + c_6_X_3_^2^ + c_7_X_1_X_2_ + c_8_X_2_X_3_ + c_9_X_1_X_3_(2)
where Y is the dependent variable (the response), c_0_.......c_9_ are the coefficients of the regression equation and X_1_, X_2_ and X_3_ are the coded values of the variables [23].

The coefficients of the regression equation were achieved by entering the experimental data of the response variable (TIC) into a specific program for the design of experiments (Minitab Statistical Software).

Then, an analysis of variance (ANOVA) was applied to determine the adequacy of the model and the significance of the coefficients. The values of the Fisher coefficient and the *p* values of statistical significance (*p* < 0.5) were evaluated in order to draw conclusions regarding the model built on the basis of experimental data.

### 3.4. Determination of the Total Iridoids Content in the Extracts

The TIC in the obtained extracts was determined by the colorimetric method using the Trim–Hill reagent [24]. The measurements were performed in triplicate using a T80 + UV–Vis spectrophotometer (PG Instruments, Lutterworth, UK). The Trim– Hill reagent was prepared by mixing glacial acetic acid, copper (II) sulfate (0.2%) and concentrated hydrochloric acid in the volumetric ratio of 10:1:0.5. The absorbance at 609 nm of the samples consisting of 100 µL of suitable diluted extract and 1 mL of Trim–Hill reagent was monitored. In order to complete the reaction between the reagent and the sample, the mixture was heated in a water bath at 100 °C for five minutes. A blank sample composed of 100 µL of diluted extract in 1 mL of water was heated under the same conditions. The TIC was obtained based on a calibration curve produced in the same conditions using aucubin as a standard. The results were expressed in µg aucubin/µL of sample.

### 3.5. Chromatographic Secondary Metabolites Profiling

Iridoids and polyphenols were the classes of secondary metabolites used to obtain the profiles of the analyzed plants by TLC and UPLC.

#### 3.5.1. Thin-Layer Chromatography (TLC)

First, TLC secondary metabolites profiles were obtained by separation on two types of chromatographic plates, one with normal phase—HPTLC silica gel 60 F_254_ of 20 × 10 cm, and the other with reverse phase—TLC silica gel 60 RP-18 F_254_ of 20× 10 cm (Merck—Darmstadt. Germany). The mobile phases were: ethyl acetate–methanol–distilled water (7.7:1.5:0.8 *v*/*v*/*v*) for the normal phase separation [25], and formic acid (0.1%)–acetonitrile (5:5 *v*/*v*) [26] for the reverse phase separation. The plates were eluted over 80 mm distance at room temperature in a twin-through chromatographic chamber (Camag, Basel, Switzerland) pre-saturated for 30 min with the mobile phase. Aliquots of 5 µL of each standard (0.1%) and 10 µL of each sample were applied as 8 mm bands, at 1.2 cm from the lowera edge of the plate with a rate of 100 nL/s using a Linomat V applicator (Camag, Basel, Switzerland). After separations, the plates were derivatized with various reagents in order to reveal the different classes of secondary metabolites and to evaluate the antioxidant activity of the samples. To highlight the presence of iridoids in the analyzed extracts, two different derivatization reagents were used: the Ehrlich reagent and anisaldehyde solution. The reagents were prepared as follows: Ehrlich reagent—0.5 g of p-(dimethylamino)benzaldehyde were solved in 150 mL of methanol and 50 mL of concentrated HCl; and the anisaldehyde solution—170 mL of methanol was mixed with 20 mL of acetic acid and 10 mL of sulfuric acid, over which 1 mL of anisaldehyde was added [27]. The eluted and dried chromatographic plates were separately sprayed with each reagent and heated for 5 min at 100 °C. The iridoids were detected as blue-grey spots on a pink background in the case of the Ehrlich reagent and blue-violet on a yellow background in the case of anisaldehyde.

For the qualitative analysis of the polyphenols from the extracts separated on the chromatographic plates, nine standards were used (flavonoids and phenolic acids), namely p-coumaric acid, vanillic acid, rutin, caffeic acid, quercetin, ferulic acid, gallic acid, chlorogenic acid and protocatechuic acid. NP/PEG derivatization reagents were used to enhance the fluorescence of compounds separated on chromatographic plates. Solutions of these reagents were prepared as follows: NP solution—1 g of 2-aminoethyldiphenylborinate was dissolved in 200 mL of ethyl acetate; PEG—10 g polyethylene glycol was dissolved in 200 mL dichloromethane. After heating the plate for 3 min at 100 °C, the plate was sprayed with NP while was hot, followed after drying by PEG [18]. Detection was performed under UV light (254 and 366 nm) before and after derivatization with NP/PEG solutions.

To evaluate the antioxidant activity of the extracts, two radicals were used: DPPH and ABTS. The DPPH• solution was prepared in ethanol at a concentration of 0.03%. The ABTS•^+^ was prepared by mixing equal volumes of a 7 mM ABTS solution with a 2.45 mM K_2_S_2_O_8_ solution and then incubating in the dark at room temperature for 16 h. Before the immersion of the chromatographic plates, the obtained ABTS•^+^ solution was diluted in a volumetric ratio of 1:1 (*v*/*v*) with ethanol. The chromatographic plates were dipped in the radical solutions using an immersion device (Camag, Basel, Switzerland) with a vertical speed of 35 mm/s and an immersion time of 3 s. The antioxidant activities were determined from the areas of the yellow and purple spots, respectively, according to the method of Hosu et al. [18].

In all cases, the plates were documented with a TLC visualizer device (Digistore 2—Camag, Basel, Switzerland) and the images were stored as JPEG files. WinCats software controlled all of the Camag instruments. The images of the plates were digitized by ImageJ (1.53t version) in order to be used for statistical analysis.

#### 3.5.2. Analysis of Extracts by Liquid Chromatography

The secondary metabolites profiles of *L. album* and *U. dioica* were also obtained by the UPLC method. The analyses were performed on an ACQUITY Arc System equipped with a quaternary pump model, Arc Quaternary; autosampler with temperature control of the sample compartment, Arc Sample Manager FTN; and a detector UV/VIS model TUV 2498 (Waters, Milford, CT, USA). An XBridgeC18 column (50 × 4.6 mm; particle size 3.5 µm) and two solvents, solvent A (0.1% formic acid in water (*v*/*v*)) and solvent B (0.1% formic acid in acetonitrile (*v*/*v*)) as mobile phases, were used for separation. The applied gradient solvent profile was as follows: 0–7.6 min, 10–19% B; 7.6–16.6 min, 19–21% B; 16.6–22.5 min, 21–40% B; and then return to the initial value 22.5–27 min, 40–10% B. The flow rate was 0.7 mL/min and the column temperature was kept constant at 25 °C. The chromatograms were recorded in UV at 210 nm, a wavelength that is specific to iridoids [22]. Aliquots of 10 µL from diluted extracts with acetonitrile (1:1, *v*/*v*) were injected.

### 3.6. Multivariate Analysis of Chromatographic Data

The unsupervised pattern recognition technique known as principal component analysis (PCA) is widely employed in the identification and discrimination of medicinal plant raw materials. Typically, in this method, the first two PCs capture most of the variations in the data and are the most useful components of data analysis. Thus, by plotting the first two PCs in the PCA plot, we could reveal the samples’ grouping and similarities and differences between samples. The closer the PC values, the higher the similarity between the samples. Using the PCs with an eigenvalue >1 and a Varimax rotation approach, hidden information can be extracted to differentiate the analyzed samples. The PCA approach was applied in this paper for the analysis of data from HPLC chromatograms (1615 variables for each sample), iridoids TLC profiles (1137 variables for each sample) and polyphenols TLC profiles (1686 variables for each sample) of *Lamium album* and *Urtica dioica* samples. For PCA analysis, the Statistica12 Software Package (StatSoft Inc. 1984–2014, USA) was used.

## 4. Conclusions

An ultrasound-assisted enzymatic extraction method for iridoids extraction was developed and was optimized by the response surface methodology and Box–Benhken model, the results proving the effectiveness of this extraction.

The results of phytochemical analysis illustrate that the two species of nettle can be differentiated based on the secondary metabolites profiles of the extracts obtained by thin-layer chromatography using both normal and reverse phases, by RP-UPLC and by the antioxidant activity evaluation with DPPH and ABTS^+^ radicals.

A clear difference in the chemical composition of the *Lamium album* and *Urtica dioica* samples was revealed using the PCA method to analyze data from the HPLC chromatogram, TLC fingerprints of iridoids and TLC fingerprints of polyphenols, respectively. In all cases, the differences between these two species are associated with the first PC (PC1) that accounts for the highest variability from the chromatographic data. Samples from *Lamium album* species collected from controlled growth conditions (from a specialized producer for medicinal plant material) have different characteristics including a different total iridoids content, different polyphenols profiles and antioxidant activity compared with the samples collected from spontaneous flora. The significant differences between these samples were revealed for the first time using the HPLC data combined with data from the TLC fingerprints of separated iridoids and polyphenols, respectively.

It has been demonstrated that a combination of chromatographic fingerprints and chemometric methods is a fast, precise and reliable method, suitable for the differentiation of similar plants. Based on this approach, this work succeeds for the first time in differentiating between *L. album* and *U. dioica*.

The obtained results revealed significant differences between the two analyzed plants, both in terms of the contained phytochemical compounds, as well as the antioxidant activity, confirming the fact that *Lamium album* has a high potential to be used in phytomedicine.

## Figures and Tables

**Figure 1 plants-12-03233-f001:**
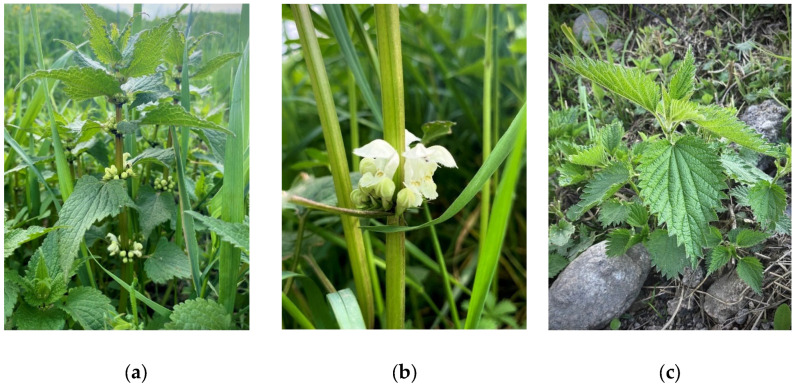
*Lamium album* (**a**) and (**b**), *Urtica dioica* (**c**).

**Figure 2 plants-12-03233-f002:**
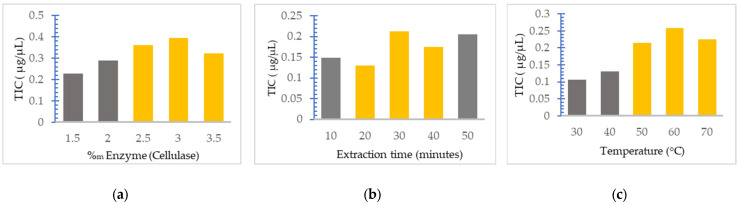
The influence of various parameters on the TIC extraction: (**a**) cellulase concentration, (**b**) extraction time, (**c**) temperature. The yellow bars marked values of the parameters were used for the method optimization, whereas the grey bars marked values of the parameters were not used for the method optimization.

**Figure 3 plants-12-03233-f003:**
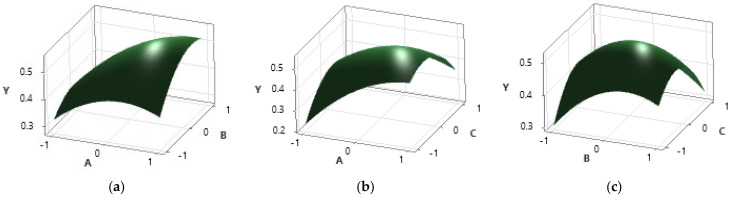
Graphical representation of the dependence of the response variable TIC (Y) on the input variables: (**a**) time (A) and enzyme concentration (B); (**b**) time (A) and temperature (C); (**c**) enzyme concentration (B) and temperature (C).

**Figure 4 plants-12-03233-f004:**
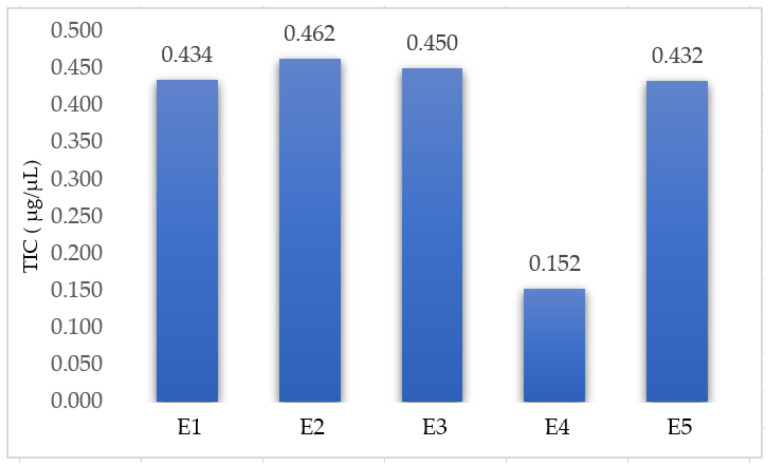
Total content of iridoids of the extracts: E1—*L. album* from Satu Mare; E2—*L. album* from Cluj-Napoca; E3—*L. album* from Dej; E4—*U. dioica* tea; and E5—*L. album* tea.

**Figure 5 plants-12-03233-f005:**
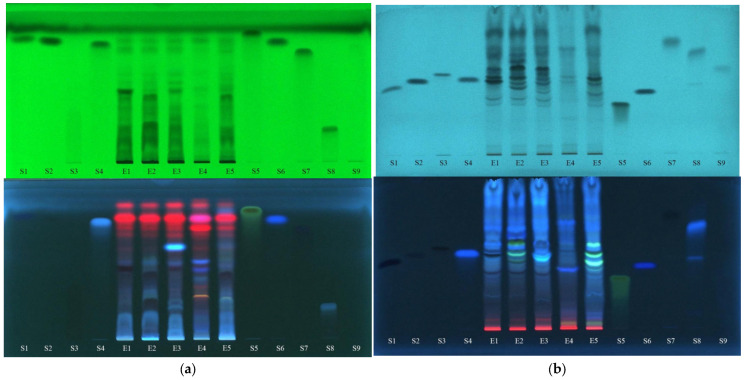
The images of chromatographic plates on the normal phase (**a**) and reverse phase (**b**) obtained before derivatization in UV light at 254 nm and 366 nm. E1—*L. album* from Satu Mare; E2—*L. album* from Cluj-Napoca; E3—*L. album* from Dej; E4—*U. dioica* tea; E5—*L. album* tea. S1—p-coumaric acid; S2—vanillic acid; S3—rutin; S4—caffeic acid; S5—quercetin; S6—ferulic acid; S7—gallic acid; S8—chlorogenic acid; S9—protocatechuic acid.

**Figure 6 plants-12-03233-f006:**
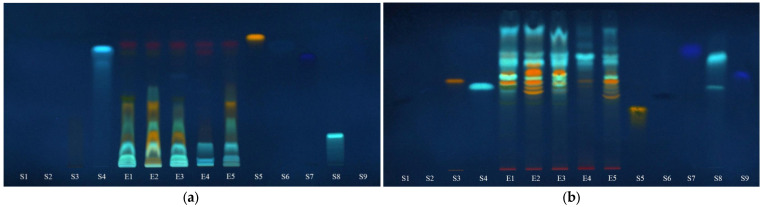
The images of chromatographic plates on normal phase (**a**) and reversed-phase (**b**) obtained after derivatization with NP/PEG in UV light at 366 nm. E1—*L. album* from Satu Mare; E2—*L. album* from Cluj-Napoca; E3—*L. album* from Dej; E4—*U. dioica* tea; E5—*L. album* tea. S1—p-coumaric acid; S2—vanillic acid; S3—rutin; S4—caffeic acid; S5—quercetin; S6—ferulic acid; S7—gallic acid; S8—chlorogenic acid; S9—protocatechuic acid.

**Figure 7 plants-12-03233-f007:**
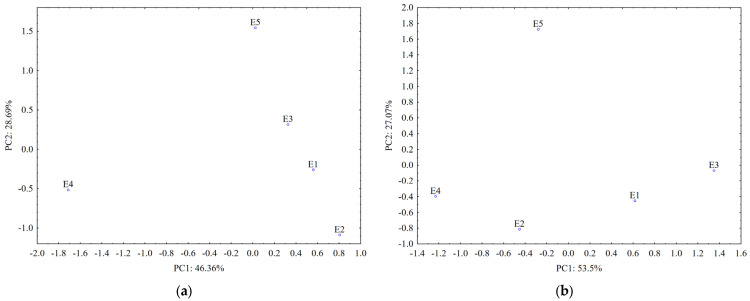
PCA classification of the analyzed samples based on data from TLC profiles of polyphenols on: (**a**) silica gel phase and (**b**) RP-18 phase (E1—*L. album* from Satu Mare; E2—*L. album* fromCluj-Napoca; E3—*L. album* from Dej; E4—*U. dioica* tea; E5—*L. album* tea).

**Figure 8 plants-12-03233-f008:**
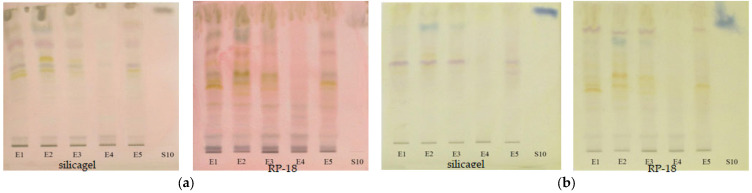
Chromatographic plates in visible light, after spraying with: (**a**)—Ehrlich reagent and (**b**)—anisaldehyde. E1—*L. album* from Satu Mare; E2—*L. album* from Cluj-Napoca; E3—*L. album* from Dej; E4—*U. dioica* tea; E5—*L. album* tea; and S10—aucubin.

**Figure 9 plants-12-03233-f009:**
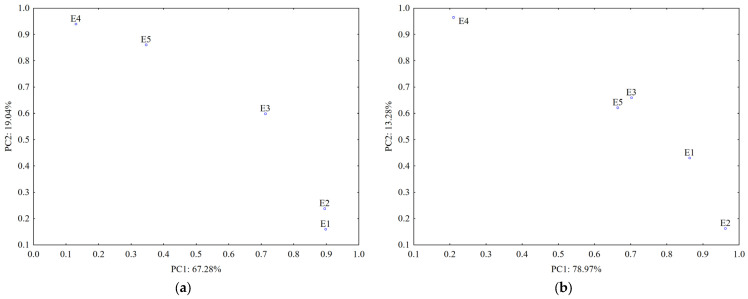
PCA classification of the analyzed samples based on data from TLC profiles of iridoids on: (**a**) silica gel plates and (**b**) RP-18 plates. (E1—*L. album* from Satu Mare; E2—*L. album* from Cluj-Napoca; E3—*L. album* from Dej; E4—*U. dioica* tea; E5—*L. album* tea).

**Figure 10 plants-12-03233-f010:**
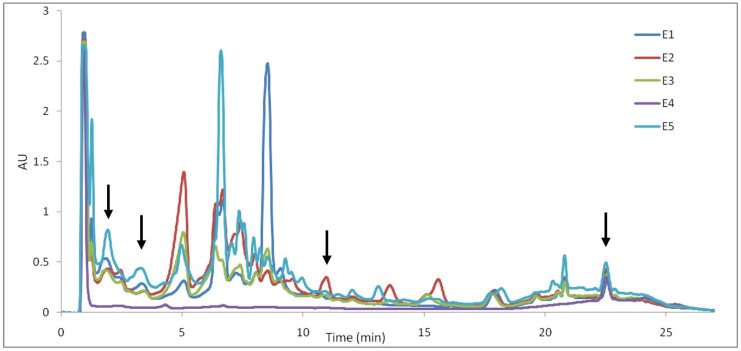
The UHPLC profiles of samples recorded at 210 nm (E1—*L. album* from Satu Mare; E2—*L. album* from Cluj-Napoca; E3—*L. album* from Dej; E4—*U. dioica* tea; E5—*L. album* tea).

**Figure 11 plants-12-03233-f011:**
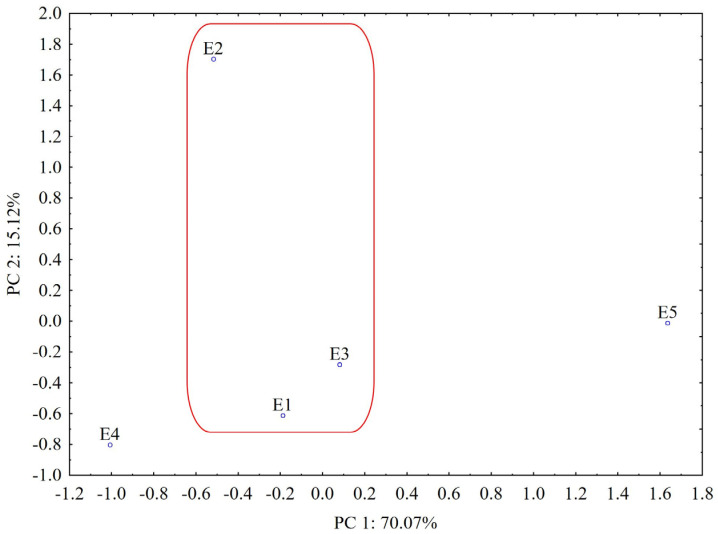
PCA classification of the analyzed samples based on UHPLC data (E1—*L. album* from Satu Mare; E2—*L. album* from Cluj-Napoca; E3—*L. album* from Dej; E4—*U. dioica* tea; E5—*L. album* tea). The grouped *L. album* samples (E1–E3) collected from spontaneous flora in Romania are highlighted.

**Figure 12 plants-12-03233-f012:**
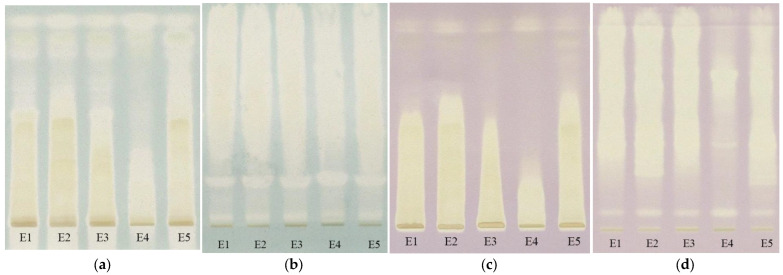
The images of the TLC plates after immersion on ABTS•^+^ ((**a**)—silicagel, (**b**)—RP-18) and DPPH ((**c**)—silicagel, (**d**)—RP-18) (E1—*L. album* from Satu Mare; E2—*L. album* from Cluj-Napoca; E3—*L. album* from Dej; E4—*U. dioica* tea; E5—*L. album* tea).

**Table 1 plants-12-03233-t001:** Level scaling of variables.

Level	A—Time (min)	B—Enzyme Concentration (%m)	C—Temperature (°C)
−1	20	2.5	50
0	30	3	60
+1	40	3.5	70

**Table 2 plants-12-03233-t002:** The Box–Behnken model with the dependent variable.

Experiment	A—Time (min)	B—Enzyme Concentration (%m)	C—Temperature (°C)	Y—TIC (µg/µL)
1	−1	−1	0	0.328
2	+1	−1	0	0.384
3	−1	+1	0	0.294
4	+1	+1	0	0.511
5	−1	0	−1	0.237
6	+1	0	−1	0.530
7	−1	0	+1	0.306
8	+1	0	+1	0.346
9	0	−1	−1	0.387
10	0	−1	−1	0.397
11	0	+1	+1	0.362
12	0	+1	+1	0.336
13	0	0	0	0.516
14	0	0	0	0.522
15	0	0	0	0.511

**Table 3 plants-12-03233-t003:** Analysis of variance for proposed regression model of response.

Source	DF*	F*-Value	Fcrit*-Value	*p**-Value
Model	9	30.02	2.6458	0.001
Linear	3	32.99		0.001
A	1	88.82	4.6001	0.000
B	1	7.60	4.6001	0.040
C	1	2.57	4.6001	0.170
Square	3	39.56		0.001
A^2^	1	29.42	4.6001	0.003
B^2^	1	38.23	4.6001	0.002
C^2^	1	68.20	4.6001	0.000
2-Way interaction	3	17.49		0.004
AB	1	12.58	4.6001	0.016
AC	1	30.99	4.6001	0.003
BC	1	8.91	4.6001	0.031
Error	5			
Total	14			

* DF—the degrees of freedom; F—the Fisher coefficient; Fcrit—the critical value of the Fisher coefficient; and *p*—the statistical significance threshold.

**Table 4 plants-12-03233-t004:** The antioxidant activities of the *L. album* and *U. dioica* extracts estimated as sum of all peaks area (arbitrary units).

Extract	DPPH•		ABTS•^+^	
	Silica Gel	RP-18	Silica Gel	RP-18
E1—*L. album* from Satu Mare	4838	9017	5618	6447
E2—*L. album* from Cluj-Napoca	5363	9123	5107	7228
E3—*L. album* from Dej	3706	9029	5398	6897
E4—*U. dioica* tea	3587	6091	5235	6345
E5—*L. album* tea	6364	8891	6316	6126

## Data Availability

The data presented in this study are available in the article and the Appendix A.

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
