# Peer review of "Use of Secondary Metabolites Profiling and Antioxidant Activity to Unravel the Differences between Two Species of Nettle"

_plants, 2023, doi:10.3390/plants12183233_

Round 1
Reviewer 1 Report
An ultrasound-assisted enzymatic extraction method for iridoids extraction was developed. The authors believe that the results of phytochemical analysis illustrate that the two species of nettle Lamium album and Urtica dioica can be differentiate based on secondary metabolites profiles of the extracts obtained by thin-layer chromatography using both normal and reverse phases, by RP-UHPLC and by the anti- oxidant activity evaluation with DPPH and ABTS+ radicals. A clearly difference in the chemical composition of the Lamium album and Urtica dioica samples were revealed using PCA method to analyse data from HPLC chromatogram, TLC fingerprints of iridoid components and TLC fingerprints of polyphenols respectively.
Unfortunately, the obtained experimental data in this work (UHPLC, TLC) does not allow us to perform chemical identification of iridoids and polyphenolic compounds. The authors use neither any standard compounds for identification of chemical constituents in the samples of the Lamium album and Urtica dioica nor HPLC-MS techniques. The peaks on the HPLC profiles (Fig. 10) were not identified. This is why the scientific soundness this manuscript is quite low.
Author Response
Indeed, the obtained data do not allow us to identify the secondary metabolites, but that was not the purpose of this work either. In the case of the TLC analysis, standards of polyphenols were used, and in the case of the UPLC analysis, the presence of some iridoids was highlighted based on the retention times obtained under the same conditions by other authors, as is also mentioned in the manuscript.
The purpose of the research was to obtain chromatographic fingerprints of the two plants that could be used to distinguish them. Chromatographic fingerprints can differentiate the samples without the need to identify any compound, something proven by other previous research, both by us and by other authors, which proves the scientific soundness of this manuscript. This was mentioned in the manuscript.
Reviewer 2 Report
Title: Use of secondary metabolites profiling and antioxidant activity to unravel the differences between two species of nettle.
Summary:
This study delves into the differentiation between two species of nettle based on secondary metabolites profiling and antioxidant activity. For clarity, enhancement, and concise presentation, the following areas of improvement are identified:
Introduction:
- Begin with broad, engaging statements.
- Avoid redundancy and improve the flow between topics.
- Clarify ambiguous statements.
- Simplify complex terms for a broader audience.
- identify the research gap and concisely state the study's objective.
- Improve the text's structure with appropriate formatting and ensure proper citations.
- Adopt a mix of active and passive voice for better readability.
Results and Discussion:
- Present results in a sequential, connected manner.
- Enhance figure captions for better data interpretation.
- Directly contrast various methods and their results.
- Delve deeper into the implications of findings and relate them to existing literature.
- Avoid repetition and improve data interpretation clarity.
- Ensure each result discussed has clear significance.
Materials and Methods:
- list chemical reagents, purity, and source.
- Specify which plant parts were used and detail the drying process.
- Elaborate on fixed parameters in extraction optimization.
- Detail chromatographic profiling methods and conditions.
- Ensure consistency in terminology, correct typographical errors, and cite references.
- Emphasize safety when handling hazardous chemicals.
In essence, while the study is comprehensive, it would gain from clearer interpretations, a streamlined presentation, and elaborated methodologies.

Minor editing of English language required
Author Response
All suggestions and comments were taken into account and the appropriate changes were made, except for the one related to enhancement of figure captions for better data interpretation. In our opinion the figure captions are suitable, but any suggestion are welcome.
Round 2
Reviewer 1 Report
Unfortunately, we still think that it is not possible to obtain ''chromatographic fingerprints'' of the two plants without identification of chemical compounds in the extracts. Although the authors applied TLC analysis, using of standard compounds for HPLC identification or HPLC-MS analysis would significantly increase the quality of the manuscript.
Reviewer 2 Report
Accept in current status